# Minimally Invasive Forefoot Surgeries Using the Shannon Burr: A Comprehensive Review

**DOI:** 10.3390/diagnostics14171896

**Published:** 2024-08-29

**Authors:** Jun Young Choi, Chul Hyun Park

**Affiliations:** 1Department of Orthopedic Surgery, Inje University Ilsan Paik Hospital, Juhwa-ro 170, Ilsanseo-gu, Goyang 10380, Republic of Korea; 2Department of Orthopedic Surgery, College of Medicine, Yeungnam University, Hyeonchung-ro 170, Nam-gu, Daegu 42415, Republic of Korea; chpark77@yu.ac.kr

**Keywords:** forefoot, minimally invasive surgery, percutaneous surgery, hallux valgus, hallux rigidus, bunionette, lesser toe deformity, Shannon burr

## Abstract

Since the early 2000s, minimally invasive forefoot surgery (MIS), particularly hallux valgus correction, has significantly advanced with the introduction of the Shannon burr. However, despite numerous relevant studies being published, no comprehensive review articles have summarized MIS for various forefoot conditions. Therefore, in this comprehensive review, we examined the relevant studies about the application of MIS (excluding arthroscopy and endoscopy) for various forefoot conditions. Additionally, we discuss the essential considerations for achieving favorable surgical outcomes and preventing complications associated with each technique. We analyzed the characteristics of each surgical procedure and identified areas for future focus. Effective surgical treatment not only requires MIS, but also the appropriate selection of patients based on suitable indications and executing procedures within the surgeon’s capabilities. We hope that this review will help readers to enhance their expertise in this field.

## 1. Introduction

Historically, once surgical treatment for a particular disease has been established and its efficacy has been demonstrated, the goal of surgeons has been to refine the procedure to minimize patient trauma. In this context, minimally invasive surgery (MIS) has become the prevailing approach in nearly all surgical fields. MIS has the advantages of reducing wound complications and promoting faster rehabilitation by minimizing surgical incisions. There is a trend toward MIS for various orthopedic conditions of the forefoot, posing a challenge to orthopedic surgeons performing foot surgeries. In our opinion, MIS in the field of foot surgery can be broadly categorized into two main concepts: the traditional concept of using arthroscopy or endoscopy to address intra- or extra-articular or tendon pathology, and the more recent concept of performing various bone and joint surgeries using the Shannon burr (Figure 1), which has gained worldwide popularity since the early 2000s. Shannon burrs were originally used by maxillofacial surgeons and dentists for the excavation, perforation, and incision of bone fragments. However, they have since been applied in hallux valgus surgeries by podiatrists and have proven useful in various areas of foot surgery. The Shannon burr is available in various diameters, lengths, and shapes, allowing for its selection based on the type and size of the bone, as well as the shape, location, and direction of osteotomy.

Following a literature search of numerous relevant studies, we found no comprehensive review article summarizing MIS for various forefoot conditions. Therefore, in this study, we aimed to exclude arthroscopic and endoscopic surgeries and conduct an in-depth review of bone and joint surgeries for deformity correction in the context of forefoot MIS. Through this review, we aimed to explore the current application of MIS techniques for various forefoot conditions, identify key considerations for achieving optimal surgical outcomes, discuss anatomical considerations for preventing complications, and propose future recommendations for researchers in the field.

## 2. Hallux Valgus (Metatarsal Osteotomy)

### 2.1. Evolution of the Technique

Hallux valgus deformity is a static subluxation of the first metatarsophalangeal (MTP) joint that primarily affects middle-aged and older women [1]. The hallux valgus is characterized by the lateral deviation of the great toe and medial deviation of the first metatarsal, resulting in the medial prominence of the first metatarsal head and the formation of a painful bunion [2]. The significance of hallux valgus deformity in the development of foot MIS is substantial. The evolution of MIS in the foot has closely followed the trajectory of advances in correcting hallux valgus deformity. Several studies on the development of surgical techniques and instruments have been published, and various research efforts are currently underway.

First-generation MIS for hallux valgus deformity correction dates back to Isham’s introduction of an incomplete, oblique extra-articular osteotomy of the first metatarsal head [3]. This method, a modification of the technique introduced by Reverdin [4] in the late 19th century, involves performing a dorsomedial closing wedge osteotomy on the first metatarsal head, followed by closing the osteotomy site without fixation (Figure 2A). While this technique has the advantage of correcting distal metatarsal articular surface deviation, it also has significant drawbacks, including the practical shortening of the metatarsal by 4–6 mm, instability at the osteotomy site leading to displacement or angulation, and a lack of effectiveness in correcting the first to second intermetatarsal angle (IMA) [5,6].

Second-generation MIS was introduced by Bösch et al. [7,8] and is characterized by performing a transverse subcapital osteotomy on the first metatarsal neck, excluding any soft tissue procedures, and stabilizing the osteotomy site with an axial Kirschner wire (K-wire) after laterally translating the metatarsal head (Figure 2B). This technique was widely used until the advent of third-generation MIS, and it is still used by many surgeons [5,9,10,11]. However, several studies have reported risks associated with this method, including malunion due to insufficient fixation, recurrence, pin-site infection, and deep infection [5,12,13,14]. Additionally, owing to the nature of transverse osteotomies, valgus angulation of the distal fragment cannot be performed, limiting the correction to the lateral translation component. Methods using intramedullary fixation devices have been developed to address the fixation issues associated with second-generation MIS [15,16]. However, these techniques have not been widely adopted because they overlap with the emergence of third-generation MIS techniques.

The new MIS method for the correction of hallux valgus deformities began with the application of the Shannon burr. This technique, known as minimally invasive chevron Akin osteotomy (MICA) or percutaneous chevron Akin osteotomy, was initially described by Vernois and Redfern [17,18,19,20,21]. The procedure involves a stab incision and an extra-articular chevron-shaped osteotomy in the distal metatarsal area (Figure 2C), allowing for three-dimensional correction through lateral translation, valgus angulation, and the slight supination of the distal fragment. Unlike the first and second generations, this method uses screws to secure the osteotomy site, thereby ensuring robust stability.

### 2.2. Surgical Outcome

Numerous researchers have reported excellent surgical outcomes using the MICA technique [12,22,23,24,25,26,27,28,29]. Lewis et al. [22] followed up 292 cases for over 2 years and reported successful results with minimal complications; a follow-up of >5 years showed a recurrence rate of 7.7% and a complication rate of 4.8%, indicating satisfactory outcomes [23]. Comparisons with conventional open surgery, primarily with scarf-Akin osteotomy [12,27,28,29], revealed that MICA osteotomies achieved excellent outcomes in all studies. A recent surge in related studies has led to a systematic review [30], suggesting that MICA osteotomy is a safe and effective surgical method. This is a promising development, especially compared to systematic reviews of techniques before third-generation MIS, which could not reach definitive conclusions about their efficacy [31,32,33].

### 2.3. Technical Considerations during Surgery

Compared with traditional chevron osteotomy, the MICA procedure involves an extracapsular osteotomy located near the first metatarsal neck. This extra-articular extracapsular osteotomy minimizes capsular trauma by preserving the proximal attachment of the capsule [34]. It also allows for greater translation and rotational correction because the mobility of the capital fragment is not constrained by capsular tightness [35]. For metataral osteotomy, a Ø2.2 × 20 mm Shannon burr is used in general. During the osteotomy procedure, continuous cooling irrigation is needed to prevent thermal damage, which can lead to local bone necrosis and compromised wound healing. After correcting the deformity, the osteotomy site is fixed with two Ø3.0 or Ø3.5 mm cannulated screws. During screw fixation, it is crucial that at least one screw is passed through the proximal fragment bicortically [28,36] (Figure 3). However, the smaller the degree of deformity correction, the more technically challenging the bicortical fixation. Recent studies have shown that transverse-shaped osteotomy, similar to osteotomies that use the second-generation MIS method, is advantageous as it achieves bicortical fixation more effectively and securely [36,37]. Lewis et al. [36] argued that transverse osteotomies preserve cortical bone at the distal level of the far cortex, reducing the risk of the screw insertion angle becoming too parallel to the skin and enabling more effective and stable screw insertion. Yoon et al. [37] also reported that the outcomes of transverse-shaped and chevron osteotomies are similar in the correction of moderate-to-severe hallux valgus deformities using third-generation MIS. Additionally, they noted that transverse osteotomy allows for easier achievement of supination in the distal fragment. It is generally believed that the valgus angulation of the distal fragment is challenging with transverse osteotomy. However, in practical surgical procedures, using a Shannon burr for transverse osteotomy creates a slightly larger gap than using a saw. Consequently, even with transverse osteotomy, a substantial degree of valgus angulation can be achieved in the distal fragment. One critical aspect of reducing postoperative osteotomy pain is smoothing the bump on the medial aspect of the proximal fragment’s osteotomy site [38]. Among the soft tissue procedures performed alongside osteotomies, adductor tenotomy via a stab incision under fluoroscopic guidance is the most commonly performed procedure [39,40].

For Akin osteotomy, a Ø2.0 × 12 mm Shannon burr is used to perform the osteotomy in the metaphyseal area of the proximal phalanx. Peiffer et al. [41] reported that a second pass with the burr during MIS Akin osteotomies results in an average medial wedge resection of 2.9 mm in the first phalanx, emphasizing that excessive burring may increase the risk of overcorrection. Therefore, unlike conventional open Akin osteotomy, which typically involves a medial closed-wedge osteotomy, the MICA procedure often employs a lateral-opening wedge [40]. After the correction of the deformity, fixation can be achieved using various methods, such as K-wires or Ø2.0 or Ø3.0 mm cannulated screws.

### 2.4. Current Issues

Some surgeons still question whether the MICA procedure, which is an osteotomy performed at the distal metatarsal level, can effectively correct severe hallux valgus with a hallux valgus angle > 40° and an IMA > 16°. Nevertheless, most published studies to date have demonstrated that MICA procedures are an effective method for correcting moderate-to-severe hallux valgus deformities [42,43,44,45,46]. In contrast, Choi et al. [47,48] argued that performing MICA osteotomies at the proximal level is more beneficial for moderate-to-severe hallux valgus deformities than at the distal level. They explained that similar to open surgery, osteotomies at the proximal metatarsal level provide greater correctional power than those performed at the distal level.

A notable recent advancement in the MICA procedure has been the development of new guiding devices. Similar to those used for anterior cruciate ligament reconstruction in knee surgeries, these devices are commercially available. Guiding devices facilitate the maintenance of the lateral translation of the distal fragment and can simplify bicortical screw fixation of the proximal fragment, which is challenging for beginners. However, no specific research has demonstrated the superiority of these devices.

In conclusion, third-generation MIS for hallux valgus deformity, which has been used for approximately a decade, has established a standardized surgical protocol and is proving to be an effective method with consistently good outcomes. Advancements in fixation devices and surgical tools are anticipated. Proper patient selection and appropriate technique application can lead to satisfactory results, even for severe hallux valgus deformities.

## 3. Hallux Valgus (First Tarsometatarsal Arthrodesis)

In addition to metatarsal osteotomy, another surgical option for hallux valgus deformity correction is first tarsometatarsal arthrodesis, also known as the Lapidus procedure, which is attempted in a minimally invasive manner. Vieira Cardoso et al. [49] introduced an arthroscopy-assisted MIS technique for this procedure. Under fluoroscopic guidance, the cartilage was abraded using a Ø2.2 or Ø3.0 mm Shannon burr or a thicker wedge burr of Ø3.1 mm or more, and the joint was fixed with cannulated screws after correcting the deformity. Instead of cannulated screw fixation, Chaparro et al. [50] reported good outcomes with axial nail fixation. Whether adequate joint preparation can be achieved is a critical consideration for surgeons performing the Lapidus procedure via MIS. A cadaveric study by Schilde et al. [51] found that while MIS resulted in 15% less cartilage denudation than open techniques, damage to the surrounding tendinous structures was minimal, suggesting that MIS could still be a favorable option. Another study [49] noted that although MIS Lapidus procedures have a lower nonunion rate and fewer wound complications than open Lapidus procedures, the degree of hallux valgus correction may be lower, warranting caution. The MIS Lapidus procedure has not been as extensively studied as the MICA procedure, with most studies limited to case series and cadaveric studies. Therefore, careful consideration is advised when selecting this procedure.

## 4. Hallux Rigidus

### 4.1. Minimally Invasive Cheilectomy

Hallux rigidus is characterized by stiffness, pain, and deformity caused by degenerative changes in the first MTP joint. It is the second most common pathological condition affecting the first ray following hallux valgus. A hallmark symptom is pain experienced during dorsiflexion of the first MTP joint. The type of surgical procedure is based on the extent of degenerative changes.

Among the various surgical options, cheilectomy, which involves the removal of bony spurs, is the primary option, sometimes in conjunction with metatarsal or proximal phalangeal osteotomy, for patients with mild to moderate degenerative changes. MIS options with favorable outcomes have been reported by various authors. Teoh et al. [52] reported a significant reduction in the Manchester–Oxford Foot Questionnaire score after minimally invasive dorsal cheilectomy (MIDC) using a Ø3.1 mm wedge burr in 87 patients with symptomatic hallux rigidus. Morgan et al. [53] compared open cheilectomy with MIDC and reported that MIDC provided comparable improvements in foot pain, function, and social aspects. In contrast, Stevens et al. [54] compared the complications and reoperation rates between MIDC and open cheilectomy, reporting a higher, albeit statistically insignificant, rate of complications and reoperations with MIDC and advising caution among surgeons.

Some authors have combined MIDC with other techniques because an advantage of MIS is that smaller incisions make it relatively easy to combine with other procedures. Glenn et al. [55] reported favorable outcomes after performing MTP joint arthroscopy after MIDC to address intra-articular pathology. De Prado et al. [56] introduced a combined procedure involving MIDC and osteotomy of the first metatarsal head and proximal phalanx using a Shannon burr.

The risk of dorsomedial cutaneous nerve injury is a critical consideration for surgeons performing MIDCs. To prevent this, it is recommended that the nerve pathway is marked on the skin if it is palpable before surgery [54,55]. Additionally, ensuring adequate irrigation during MIDCs is crucial for preventing the retention of bone debris, which can lead to intra-articular incarceration, inflammation, and pain [55]. There have not yet been any long-term follow-up studies published on MIDC, and future research in this area is recommended.

### 4.2. Minimally Invasive First Metatarsophalangeal Arthrodesis

Arthrodesis, a joint-sacrificing procedure, including arthrodesis, interpositional arthroplasty [57], total joint arthroplasty [58], or hemiarthroplasty using polyvinyl alcohol hydrogel [59], may be considered in cases that have progressed to end-stage arthritis.

Previous studies have reported outcomes of arthroscopy-assisted first MTP arthrodesis [60,61]. Percutaneous techniques without arthroscopy have been introduced that use a wedge burr [62,63,64] or Shannon burr [65,66] for articular surface preparation, followed by cannulated screw fixation. According to a systematic review by Hodel et al. [67], the outcomes of MIS first MTP arthrodesis are comparable to those of open surgery. Angthong et al. [68] conducted a cadaveric study and concluded that minimally invasive and arthroscopic first MTP arthrodesis did not significantly differ in terms of articular surface preparation and safety profiles. Similar to MIDC, inadequate irrigation of bone debris can lead to postoperative pain, necessitating careful attention among surgeons [66]. Furthermore, surgeons should exercise caution when preventing injuries to the dorsomedial and dorsolateral cutaneous nerves.

## 5. Bunionette

A bunionette, also known as a Tailor’s bunion, is a static deformity characterized by valgus deviation of the fifth metatarsal and varus deviation of the fifth toe. This condition is often accompanied by subluxation of the fifth MTP joint. The prominent fifth metatarsal head is typically irritated by tight-fitting shoes, leading to bursitis and hyperkeratosis of the skin, which subsequently causes pain. The primary goal of surgery is to alleviate the prominence of the fifth metatarsal head either by resecting part of the head (condylectomy) or by medializing its position through an osteotomy.

The most widely recognized technique is subcapital transverse osteotomy [7], which is similar to second-generation MIS of the hallux valgus deformity. Subcapital transverse osteotomy requires a mini-incision of less than 1 cm in the fifth metatarsal neck area, followed by a transverse osteotomy at the metatarsal neck. Gianini et al. [69] named this procedure SERI (Simple, Effective, Rapid, Inexpensive) osteotomy, which remains the most widely used term. Subsequently, the fifth metatarsal head is translated medially, and a longitudinal K-wire is inserted from the toe tip into the intramedullary space of the proximal fragment. However, if the fourth to fifth IMA is excessively large, a proximal level osteotomy may be considered, although this approach is associated with a higher risk of delayed bone healing than distal osteotomy [70,71].

Given the minimal incision required for SERI osteotomy, the necessity of further reducing the incision size might be questioned. However, before using the Shannon burr, Magnan et al. [72] developed a technique using a micromotorized Lindemann bone cutter, which enabled even smaller incisions for SERI osteotomy. Lui [73] also reported satisfactory outcomes after percutaneous osteotomy at the metatarsal shaft and proximal metatarsal levels. A significant distinction of new MIS methods employing the Shannon burr is the absence of fixation. Laffenêtre et al. [74] described a technique involving condylectomy with a conical burr and distal metatarsal medial closing wedge osteotomy using a Ø2.0 × 12 mm straight Shannon burr. This approach maintains correction using dressing alone, thereby allowing bony union without additional fixation. Morawe and Schmieschek [75] reported that medial closing wedge osteotomy without fixation can also be performed at the diaphyseal and proximal metatarsal levels. Although there is concern about delayed union due to the absence of fixation, most symptoms related to this issue reportedly improve 6 months postoperatively [71].

Because research on new MIS techniques for bunionette correction is limited, surgeons need to use these methods cautiously. Ultimately, surgeons should select an appropriate surgical technique. However, having various MIS options can be beneficial, broadening the range of choices for both patients and surgeons and potentially improving outcomes.

## 6. Metatarsophalangeal Joint Problem of the Lesser Toes

The term “MTP joint problem of the lesser toes” encompasses various conditions; however, this study focuses on the MIS technique in cases of metatarsal parabola disruption. The metatarsal heads from the first to the fifth form a gentle parabola. An abnormal length of a particular metatarsal bone or when conditions such as hallux valgus deformity alter the relative length of the first metatarsal to the second can result in metatarsalgia, changes in the height of the metatarsal heads from the ground, MTP joint subluxation, and plantar skin hyperkeratosis [76]. The primary goal of surgical treatment is to restore the metatarsal parabola by shortening the long metatarsal. Among the various surgical techniques reported to date, Weil osteotomy is the most commonly used technique [77]. This procedure involves performing an oblique osteotomy from the dorsal distal aspect of the metatarsal head in the plantar proximal direction at the metatarsal neck. Following osteotomy, the distal fragment is gently pushed to achieve the desired shortening, and then one or two mini-screws are used for dorsoplantar fixation.

Distal metatarsal minimally invasive osteotomy (DMMO) is a representative MIS technique for addressing this issue. De Prado et al. [78] proposed performing an osteotomy similar to Weil osteotomy using a Ø2.0 × 12 mm Shannon burr with no fixation, allowing immediate full forefoot weight bearing. They argued that this approach enables the natural displacement of the distal fragment to the required position, leading to bony union. Although there may be concerns about nonunion and malunion at the osteotomy site due to lack of fixation, a previous study [79] reported no significant differences in surgical outcomes between procedures with and without fixation. Despite the fact that overcoming a significant learning curve is necessary to prevent osteotomy site malunion and nonunion [80], numerous studies have reported favorable outcomes following DMMO [81,82,83]. An early systematic review of DMMOs [84] expressed negative opinions regarding the procedure. However, DMMO has continued to evolve with various modifications, including changes in the osteotomy site [85] and alterations in the osteotomy direction [86]. Despite being relatively simple, without fixation performed through a minimal incision, DMMO has shown outcomes comparable to those of traditional open procedures. This suggests that DMMO is a promising area for future development, offering similar efficacy to potentially less invasive approaches.

## 7. Lesser Toe Deformity

Lesser toe deformities include mallet, hammer, claw, and curly toe deformities. MIS techniques for correcting these deformities can be categorized into osteotomy, condylectomy, tenotomy, and arthrodesis [87]. Often, these procedures require multiple phalangeal osteotomies and combined soft tissue procedures, rather than a single phalanx procedure. MIS phalangeal osteotomy involves performing an incomplete closing wedge osteotomy at the apex of the deformity using a Ø2.0 × 8 mm Shannon burr [88]. After correcting the deformity by closing the osteotomy gap, the osteotomy site is left untouched with no internal fixation, with the expectation of union with only the dressings applied. Several authors have reported good results using this method [89,90,91]. Conversely, Lee et al. [88] reported successful results in correcting symptomatic curly toe deformity by performing dorsolateral closing wedge osteotomies at the phalangeal shaft level and temporarily fixing it with a K-wire for 4 weeks. Minimally invasive tenotomy can be also performed on various tendons, including the flexor digitorum longus, brevis, and extensor digitorum [87]. However, it has been reported that incomplete tenotomy may lead to poor outcomes and requires caution [92].

It is important to be mindful of potential digital nerve damage during all procedures. Longitudinal cannulated screw fixation, which is similar to the open technique, is necessary for arthrodesis. Mateen et al. [93] reported no significant differences in surgical outcomes between open and MIS arthrodesis techniques for hammer toe correction. Although only a few relevant studies have focused on MIS for lesser toe deformity correction, considering that the union time at the phalangeal osteotomy site is shorter than that at other bones, MIS can potentially maximize its advantages in this field. Therefore, it is a promising technique for future development.

## 8. Conclusions

In this review, we examined the characteristics of each surgical procedure and discussed areas for future research by surgeons. Additionally, this review discussed essential considerations for surgeons to ensure successful outcomes and achieve more satisfactory outcomes. Furthermore, through this review, we confirmed that significant technological advancements have been made in forefoot MIS using the Shannon burr, establishing it as a highly important area in the field. And this review also confirms the immense potential for further development through ongoing research.

The key to satisfactory surgical outcomes is not merely performing MIS, but appropriately selecting patients based on suitable indications and executing the procedures within the surgeon’s capabilities. We hope that this review will help readers enhance their expertise in this field.

## Figures and Tables

**Figure 1 diagnostics-14-01896-f001:**
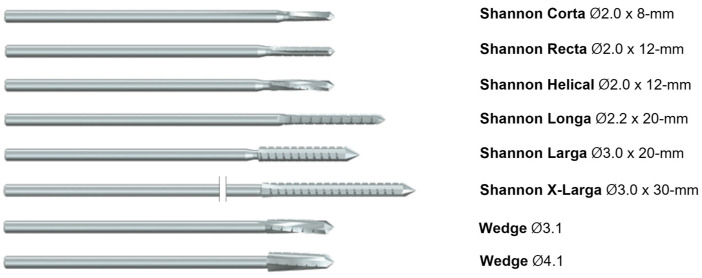
**Shannon burr set used in surgical procedures.** The set includes burrs of various sizes and types, designed for precise cutting and shaping during minimally invasive surgeries.

**Figure 2 diagnostics-14-01896-f002:**
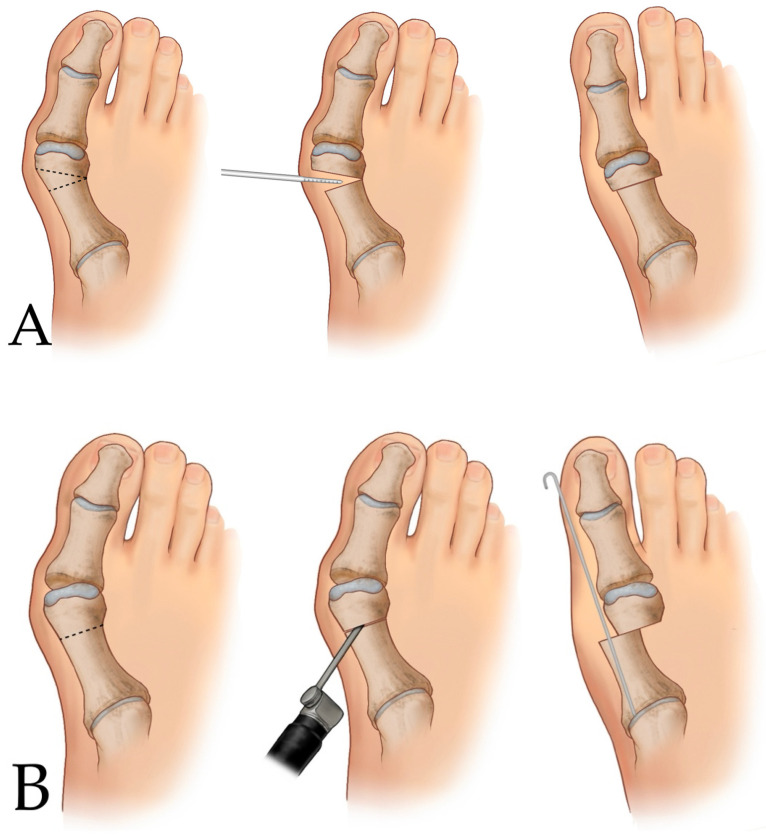
Schematic illustration of the first-, second-, and third-generations of minimally invasive surgery (MIS) for hallux valgus correction. MIS for hallux valgus correction has evolved to achieve more effective and stronger fixations with smaller incisions. First-generation MIS involves performing a medial-closing wedge osteotomy on the first metatarsal head, followed by closure of the osteotomy site with no fixation (**A**). Second-generation MIS is characterized by performing a transverse subcapital osteotomy on the first metatarsal neck, excluding any soft tissue procedures, and stabilizing the osteotomy site using an axial Kirschner wire after the lateral translation of the metatarsal head (**B**). Third-generation MIS involves an extra-articular chevron-shaped osteotomy in the distal metatarsal area using a Shannon burr (**C**), which allows for three-dimensional correction through lateral translation, valgus angulation, and the slight supination of the distal fragment.

**Figure 3 diagnostics-14-01896-f003:**
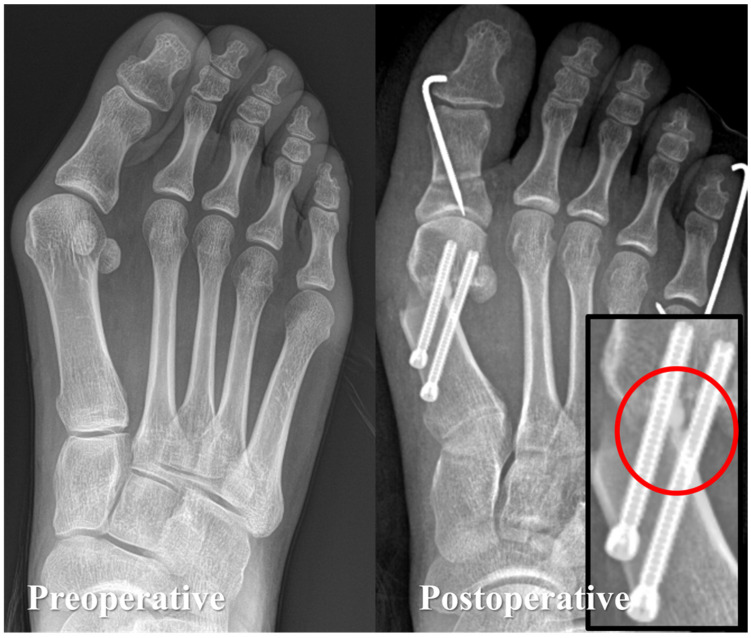
**Example of bicortical screw fixation of the proximal fragment in minimally invasive chevron Akin (MICA) osteotomies.** During the screw fixation process in MICA procedures, it is crucial that the fixation security passes at least one screw through the proximal fragment bicortically.

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
