# Peer review of "Minimally Invasive Forefoot Surgeries Using the Shannon Burr: A Comprehensive Review"

_diagnostics, 2024, doi:10.3390/diagnostics14171896_

Round 1

Reviewer 1 Report

Comments and Suggestions for Authors

Drawing wrong (saw blade for Isham, chevron too distal)

Discuss more about why chevron or why straight cut. Need more development.

Initial original references are missing

More explanation about the burr

Author Response

Reviewer 1

Comments and Suggestions for Authors

Comment 1 Drawing wrong (saw blade for Isham, chevron too distal)

Answer: Thank you for your thoughtful feedback. We have revised the illustration in Figure 2 to incorporate the feedback provided.

Comment 2 Discuss more about why chevron or why straight cut. Need more development.

Answer: Thank you for your thoughtful insights. I have strengthened the discussion on the relevant section accordingly.

Comment 3 Initial original references are missing

Answer: Thank you for your thoughtful feedback. I have reviewed and added several initial references accordingly

Comment 4 More explanation about the burr

Answer: Thank you for your thoughtful feedback. We have revised the introduction accordingly and also included considerations for the use of the Shannon burr.

Additional changes

  1. Figure 1 has been completely redesigned to incorporate the recommendations provided.
  2. We have carefully revised each of the points you highlighted in the PDF according to your requests.

Reviewer 2 Report

Comments and Suggestions for Authors

General Comments

This comprehensive review discusses the various forefoot MIS procedures for the most common forefoot anatomical deformities. It adequately covers the history and development of MIS techniques over time and then delves into more details about the indications, technique, and outcomes of each procedure citing the appropriate references.

Abstract

Appropriately brief and describes the manuscript adequately.

Introduction

Lines 23-26: It would be clearer to the reader to have another sentence bridging these two sentences. The transition is abrupt and impacts the coherence of the introduction. 

2. Hallux Valgus (Metatarsal Osteotomy)

Line 55: High contact pressure where? Please be more specific in this statement. 

Line 59-60: The word numerous has been used twice, and I recommend rephrasing the sentence for the sake of clarity.

Lines 144-145: Why is valgus angulation less favorable? Please elaborate more on this statement using evidence from the literature.

Lines 167-172: It is also worth discussing the degree of correction in MIS Akin osteotomy following several Shannon burr passes. I suggest you refer to and cite the following paper describing the degree of correction in MIS Akin and dorsiflexion osteotomies to inform the reader about the degree of correction following a set number of burr passes: https://doi.org/10.1016/j.fas.2023.12.008

7. Lesser Toe Deformity

Line 323: Please cite a reference from the literature to support this statement and this burr caliber.

8. Conclusion

I believe the conclusion does not sufficiently address the content of the manuscript. Instead of mentioning what was "examined" or "covered," provide a general summary of the text, focusing on and emphasizing the importance of MIS surgery and briefly mentioning the main points related to the aims of this comprehensive review.

Comments on the Quality of English Language

The language of the introduction requires extensive edits to improve readability and coherence. Further elaboration may help the introduction set the stage for the review and transition to the aims. Otherwise, the bulk of the review text seems grammatically sound, clearly and reliably describing the procedures in a systematic way. The conclusion requires extensive rewriting, as noted in the comments.

Author Response

Reviewer 2

Comments and Suggestions for Authors

General Comments

This comprehensive review discusses the various forefoot MIS procedures for the most common forefoot anatomical deformities. It adequately covers the history and development of MIS techniques over time and then delves into more details about the indications, technique, and outcomes of each procedure citing the appropriate references.

Answer: We sincerely appreciate your positive evaluation of our manuscript.

Abstract

Appropriately brief and describes the manuscript adequately.

Answer: We sincerely appreciate your positive evaluation of our manuscript.

Introduction

Lines 23-26: It would be clearer to the reader to have another sentence bridging these two sentences. The transition is abrupt and impacts the coherence of the introduction. 

Answer: Thank you for your precise feedback. I have revised the introduction as per your suggestions.

  1. Hallux Valgus (Metatarsal Osteotomy)

Line 55: High contact pressure where? Please be more specific in this statement. 

Answer: Thank you for your thoughtful feedback. We have made the necessary revisions to that section.

Line 59-60: The word numerous has been used twice, and I recommend rephrasing the sentence for the sake of clarity.

Answer: Thank you for your thoughtful feedback. We have revised that section accordingly.

Lines 144-145: Why is valgus angulation less favorable? Please elaborate more on this statement using evidence from the literature.

Answer: Thank you for your accurate feedback. We have made substantial revisions to this section.

Lines 167-172: It is also worth discussing the degree of correction in MIS Akin osteotomy following several Shannon burr passes. I suggest you refer to and cite the following paper describing the degree of correction in MIS Akin and dorsiflexion osteotomies to inform the reader about the degree of correction following a set number of burr passes: https://doi.org/10.1016/j.fas.2023.12.008

Answer: Thank you for your thoughtful suggestion. We have added the recommended reference to my manuscript and incorporated the suggested content into the text.

  1. Lesser Toe Deformity

Line 323: Please cite a reference from the literature to support this statement and this burr caliber.

Answer: Thank you for your thoughtful feedback. I have revised the sentence accordingly and added the relevant reference.

  1. Conclusion

I believe the conclusion does not sufficiently address the content of the manuscript. Instead of mentioning what was "examined" or "covered," provide a general summary of the text, focusing on and emphasizing the importance of MIS surgery and briefly mentioning the main points related to the aims of this comprehensive review.

Answer: Thank you for your insightful feedback. In response to your suggestions, I have added new content to the conclusion section and made necessary revisions throughout the relevant parts.

Comments on the Quality of English Language

The language of the introduction requires extensive edits to improve readability and coherence. Further elaboration may help the introduction set the stage for the review and transition to the aims. Otherwise, the bulk of the review text seems grammatically sound, clearly and reliably describing the procedures in a systematic way. The conclusion requires extensive rewriting, as noted in the comments.

Answer: This manuscript has been professionally edited by an expert at Editage®, a specialized English language editing service. I will attach the English editing certificate for your reference. We paid particular attention to the introduction section.

Reviewer 3 Report

Comments and Suggestions for Authors

Thank you for your effort in the study. 

The article is generally well-written and the readers will be very interested.

Author Response

Reviewer 3

Comments and Suggestions for Authors

Thank you for your effort in the study. 

The article is generally well-written and the readers will be very interested.

Answer: Thank you for your positive evaluation of our paper.
